# Pre-Surgery Demographic, Clinical, and Symptom Characteristics Associated with Different Self-Reported Cognitive Processes in Patients with Breast Cancer

**DOI:** 10.3390/cancers14133281

**Published:** 2022-07-05

**Authors:** Yu-Yin Allemann-Su, Marcus Vetter, Helen Koechlin, Steven M. Paul, Bruce A. Cooper, Kate Oppegaard, Michelle Melisko, Jon D. Levine, Yvette Conley, Christine Miaskowski, Maria C. Katapodi

**Affiliations:** 1Department of Clinical Research, University of Basel, 4055 Basel, Switzerland; y.su@unibas.ch; 2Department of Oncology, Cantonal Hospital Basel-Land, 4410 Listel, Switzerland; marcus.vetter@ksbl.ch; 3Division of Child and Adolescent Health Psychology, Department of Psychology, University of Zurich, 8050 Zurich, Switzerland; helen.koechlin@uzh.ch; 4Division of Clinical Psychology and Psychotherapy, Faculty of Psychology, University of Basel, 4055 Basel, Switzerland; 5Department of Anaesthesiology, Critical Care and Pain Medicine, Boston Children’s Hospital, Harvard Medical School, Boston, MA 02115, USA; 6School of Nursing, University of California San Francisco, San Francisco, CA 94143, USA; steven.paul@ucsf.edu (S.M.P.); bruce.cooper@ucsf.edu (B.A.C.); kate.oppegaard@ucsf.edu (K.O.); chris.miaskowski@ucsf.edu (C.M.); 7School of Medicine, University of California San Francisco, San Francisco, CA 94143, USA; michelle.melisko@ucsf.edu (M.M.); jon.levine@ucsf.edu (J.D.L.); 8School of Nursing, University of Pittsburgh, Pittsburgh, PA 15261, USA; yconley@pitt.edu

**Keywords:** breast cancer, cancer-related cognitive impairment, cognitive flexibility, inhibitory control, working memory

## Abstract

**Simple Summary:**

One in three patients with breast cancer report cancer-related cognitive impairment (CRCI) even before treatment. CRCI can persist and negatively impact patients’ quality of life. We used a self-report measure to assess CRCI. We assessed patients’ ability to plan and solve everyday life problems, concentrate, and have meaningful relationships with others. We evaluated subgroups of patients with different profiles regarding these abilities and whether they had different demographic and clinical characteristics. Our analyses showed that 64.2%, 43.3%, and 40.1% of the patients had clinically meaningful decrements in their abilities to plan and problem-solve, concentrate, and have meaningful relationships with others, respectively, from prior to through to 6 months after surgery. Pre-surgery symptoms (i.e., anxiety, depression, fatigue, sleep disturbance) and other characteristics (e.g., lower functional status, higher comorbidity) were associated with worse CRCI profiles and may be potential targets for personalized interventions.

**Abstract:**

Cancer related cognitive impairment (CRCI) is a common and persistent symptom in breast cancer patients. The Attentional Function Index (AFI) is a self-report measure that assesses CRCI. AFI includes three subscales, namely effective action, attentional lapses, and interpersonal effectiveness, that are based on working memory, inhibitory control, and cognitive flexibility. Previously, we identified three classes of patients with distinct CRCI profiles using the AFI total scores. The purpose of this study was to expand our previous work using latent class growth analysis (LCGA), to identify distinct cognitive profiles for each of the AFI subscales in the same sample (i.e., 397 women who were assessed seven times from prior to through to 6 months following breast cancer surgery). For each subscale, parametric and non-parametric statistics were used to determine differences in demographic, clinical, and pre-surgical psychological and physical symptoms among the subgroups. Three-, four-, and two-classes were identified for the effective action, attentional lapses, and interpersonal effectiveness subscales, respectively. Across all three subscales, lower functional status, higher levels of anxiety, depression, fatigue, and sleep disturbance, and worse decrements in energy were associated with worse cognitive performance. These and other modifiable characteristics may be potential targets for personalized interventions for CRCI.

## 1. Introduction

Cancer-related cognitive impairment (CRCI) occurs in 30–35% of patients with non-central nervous system (non-CNS) cancers, even prior to initiation of treatment [1,2]. Based on findings from both self-report measures and neuropsychological tests, executive functions are impaired in patients with CRCI [3]. Working memory, inhibitory control, and cognitive flexibility are the building blocks of cognitive processes described as “executive functions” [4]. Working memory is one’s ability to hold information in mind and to mentally work with it, which is critically important for reasoning and decision-making [4,5]. Inhibitory control includes cognitive inhibition (i.e., inhibition of thoughts and memories), selective attention (i.e., ability to control one’s attention), and self-control (i.e., ability to control behavior, thoughts and/or emotions). Inhibitory control is needed to react in ways that override strong internal predispositions or external distractions. Cognitive flexibility is the ability to change perspectives and adjust to changing demands, and it builds on working memory and inhibitory control. Working memory and inhibitory control co-occur and cannot be distinguished from one another. Working memory, inhibitory control, and cognitive flexibility support reasoning, planning, problem solving, executing actions, adapting to everyday life, and managing social interactions [4,5,6,7]. 

Working memory, selective attention, and self-control can be compromised by physical and psychological illnesses [8,9,10,11]. In patients with breast cancer, deficits in working memory and selective attention were found throughout the cancer trajectory, from pre-surgery, during, and after treatment [12,13,14,15,16,17,18,19,20,21,22,23,24,25,26]. Across multiple cross-sectional [13,14,15,27] and longitudinal [12,18,26] studies, demographic and clinical characteristics associated with worse CRCI included younger age [14,15,26], lower annual income [26], higher comorbidity burden [18,26], lower functional status [26], higher symptom distress [13,14], and higher total mood disturbance [13,14]. Pre-surgical symptoms associated with worse CRCI included higher levels of anxiety [12,18,26], depression [12], fatigue [12,18], and sleep disturbance [12,18], and lower levels of energy [18]. Deficits in working memory and selective attention have significant negative consequences in that patients may not be able to actively remember detailed medical information; focus on self-care and other activities that require ignoring or blocking out distractions; adjust to the demands of the disease; and establish priorities related to treatment adherence [2,28,29]. 

In our previous studies, we used a self-reported measure of CRCI and growth mixture modeling (GMM) and we identified three subgroups of patients with breast cancer with distinct CRCI profiles [26,30]. However, we did not distinguish among the different cognitive processes that constitute “executive functions”, namely working memory, inhibitory control, and cognitive flexibility. In this analysis we want to extend our initial findings. Using the same sample of women with breast cancer and a self-reported measure of CRCI, separate latent class growth analyses (LCGAs) examined whether there are distinct patient profiles based on assessments of cognitive processes that are based on working memory, inhibitory control, and cognitive flexibility. We also examined for risk factors among demographic and clinical characteristics and pre-surgical psychological and physical symptom severity scores that can potentially inform the development of targeted interventions for CRCI [9,31].

## 2. Materials and Methods

This analysis is part of a larger, prospective, longitudinal study that evaluated multiple symptoms in patients who underwent surgery and adjuvant treatment for breast cancer [32]. Patients were recruited from breast care centers in a Comprehensive Cancer Center, two public hospitals, and four community practices in a U.S. west coast metropolitan area. Eligible patients were English-speaking women diagnosed with breast cancer; older than 18 years; scheduled to undergo surgery on one breast; and able to provide written informed consent. Patients scheduled to have surgery on both breasts and those with distant metastases at the time of diagnosis were excluded. Of the 516 patients who were approached, 410 enrolled in the study (79.5% response rate) and 397 completed the enrollment assessment. The most common reasons for refusal were being too busy or feeling overwhelmed.

### 2.1. Study Procedures

The study was approved by the Committee on Human Research at the University of California San Francisco and by the Institutional Review Boards at each of the study sites. During preoperative visits, a clinical staff member explained the study and invited patients to participate. Patients who were willing to participate were introduced to a research nurse who determined their eligibility. After providing written informed consent, patients completed the enrollment questionnaires an average of four days prior to surgery. Follow-up questionnaires were completed each month for 6 months after surgery (i.e., seven assessments over 6 months).

### 2.2. Instruments

#### 2.2.1. Demographic and Clinical Characteristics

Patients completed a demographic questionnaire, the self-reported Karnofsky Performance Status (KPS) scale [33], and the Self-administered Comorbidity Questionnaire (SCQ) [34]. Medical records were reviewed for disease and treatment information.

#### 2.2.2. Attentional Function

The Attentional Functional Index (AFI) is a self-report measure that assesses for CRCI. AFI items provide insights into deficits experienced by patients in the effective engagement in daily activities [15]. Following a factor analysis, 13 items were scored into three subscales (i.e., effective action, attentional lapses, interpersonal effectiveness) that assess cognitive processes based on the interactions among working memory, selective attention, self-control, and cognitive flexibility (Table 1). The effective action subscale assesses an individual’s ability to engage and complete purposeful actions (i.e., reasoning, planning, executing, problem-solving). The attentional lapses subscale assesses difficulties with selective attention (i.e., actively inhibiting distractions). The interpersonal effectiveness subscale assesses an individual’s ability to maintain meaningful personal relationships and to respond to lack of inhibitory control (Figure 1) [15,35].

The 13 items are scored on an 11 point numeric rating scale (NRS) that ranged from 0 (not at all well/not at all/extremely easy) to 10 (extremely well/all the time/extremely hard) [15]. Three average subscale scores (i.e., 8.08, 8.87, 7.88) were calculated, with higher scores indicating better cognitive function. While clinically meaningful cutoff scores for each subscale are not available, the cutoff scores for the total AFI are: <5 indicates low function, 5 to 7.5 indicates moderate function, and >7.5 indicate high function. In this study, Cronbach’s alphas were 0.94, 0.82, and 0.75 for the effective action, attentional lapses, and interpersonal effectiveness subscales, respectively.

#### 2.2.3. Psychological Symptoms

##### Anxiety

State and trait anxiety were assessed using the 20-item Spielberger State-Trait Anxiety Inventories (STAI-S and STAI-T, respectively) [36]. Total scores for each scale range from 20 to 80, with higher scores indicating greater anxiety. Scores of ≥31.8 and ≥32.2 suggest high levels of trait and state anxiety, respectively [37,38]. In this study, Cronbach’s alphas for the STAI-T and STAI-S were 0.88 and 0.95, respectively.

##### Depression

The 20-item Center for Epidemiologic Studies—Depression (CES-D) scale was used to assess depressive symptoms [39]. Total scores can range from 0 to 60, with scores of ≥16 indicating the need for clinical evaluation for depression. In this study, Cronbach’s alpha for the total CES-D score was 0.90.

#### 2.2.4. Physical Symptoms

##### Fatigue and Energy

The 18-item Lee Fatigue Scale (LFS) was designed to assess physical fatigue and energy [40]. Each item was rated on a 0 to 10 NRS. Total fatigue and energy scores were calculated as the mean of the 13 fatigue items and the 5 energy items, with higher scores indicating greater fatigue severity and higher levels of energy. Cutoff scores of ≥4.4 and ≤4.8 indicate clinically meaningful levels of fatigue and decrements in energy levels, respectively [41]. In this study, Cronbach’s alphas for the fatigue and energy scales were 0.96 and 0.93, respectively.

##### Sleep Disturbance 

The 21-item General Sleep Disturbance Scale (GSDS) was designed to assess sleep disturbance in the past week [42]. Each item was rated on a 0 (never) to 7 (everyday) NRS. The GSDS total score is the sum of the seven subscale scores that range from 0 (no disturbance) to 147 (extreme sleep disturbance). Higher total and subscale scores indicate higher levels of sleep disturbance. A GSDS total score of ≥43 indicates clinically meaningful levels of sleep disturbance [37]. In this study, Cronbach’s alpha for GSDS total score was 0.86.

##### Pain

Breast pain was evaluated using the Breast Symptoms Questionnaire (BSQ) [43,44]. Part 1 of the BSQ obtained information on the occurrence of pain in the affected breast. Patients who reported breast pain were asked to complete Part 2 of the BSQ, that assessed pain intensity “right now”, average daily pain, and worst pain ever, using 0 (no pain) to 10 (worst imaginable pain) NRSs; as well as number of days per week with pain and number of hours per day in pain. Patients who reported breast pain rated its level of interference using a 0 (no interference) to 10 (completed interference) NRSs. The eight items that assessed pain interference were adapted from the interference scale of the Wisconsin Brief Pain Inventory (BPI) [45].

### 2.3. Statistical Analyses

Data were analyzed using the Statistical Package for the Social Science (SPSS) version 28 (International Business Machines, Armonk, NY, USA) and Mplus version 6.11 (Muthen and Muthen, Los Angeles, CA, USA). Latent class growth analysis (LCGA) with robust maximum likelihood estimation was used to identify subgroups of patients with similar profiles for the three AFI subscales from prior to through to 6 months after surgery. The LCGA methods are described in detail elsewhere [26,46].

Descriptive statistics and frequency distributions were generated for sample characteristics and symptom scores. For each AFI subscale, parametric and non-parametric tests were used to evaluate for differences in demographic, clinical, and symptom characteristics among the classes. A *p*-value of <0.05 was considered statistically significant. Post hoc contrasts were performed using a Bonferroni corrected *p*-value of <0.017 (0.05/3 for three possible pairwise comparisons) or <0.008 (0.05/6 for six possible pairwise comparisons).

## 3. Results

### 3.1. Identification of Subgroups within Each AFI Subscale-LCGA Analyses

The fit indices for the three AFI subscales that were used to select the final class solutions are listed in Table 2. The parameter estimates for each of the identified classes are listed in Table 3. For effective action, three classes were identified (Figure 2). For attentional lapses, four classes were identified (Figure 3). For interpersonal effectiveness, two classes were identified (Figure 4).

### 3.2. Effective Action Latent Classes

For effective action, 35.8% (n = 142) of patients were in the High effective action class. These patients had estimated scores of 8.08 prior to surgery that increased slightly over 6 months. The second subgroup was named the Moderate effective action class (n = 160, 40.3%), who had estimated scores of 6.50 prior to surgery. Their mean scores decreased slightly until the 3-month assessment that was followed by slight increases over the next 3 months. The third subgroup was named the Low effective action class (n = 95, 23.9%), who had estimated scores of 4.12 prior to surgery that remained stable over 6 months. 

#### 3.2.1. Differences in Demographic and Clinical Characteristics 

As shown in Table 4, compared to the High effective action class, the Moderate class was younger, and more likely to be diagnosed with a higher breast cancer stage and to have received adjuvant chemotherapy in the 6 months after surgery. Compared to the High effective action class, the Low class was more likely to have a lower annual income, more likely to report a diagnosis of depression, and less likely to have a progesterone receptor positive tumor. Compared to the Moderate effective action class, the Low class was less likely to be employed at the time of enrollment. Compared to the other two effective action classes, the Low class had a higher comorbidity burden and lower functional status. 

#### 3.2.2. Differences in Psychological and Physical Symptoms 

For trait anxiety, state anxiety, depression, fatigue, decrements in energy, and sleep disturbance, the scores followed the same pattern (i.e., High < Moderate < Low; Table 4). For breast pain, compared to the High effective action class, a larger percentage of patients in the Moderate class reported experiencing pain in the affected breast area prior to surgery. Compared to the Moderate effective action class, the Low class reported higher scores for “pain right now”. Compared to the other two effective action classes, the Low class reported higher scores for average daily pain, worst pain, number of days per week in pain, and pain interference.

### 3.3. Attentional Lapses Latent Classes

For attentional lapses, 17.6% (n = 70) of the patients were in the Very Low level of attentional lapses class. These patients had estimated scores of 8.87 that increased slightly over 6 months. The second subgroup was named the Low level of attentional lapses class (n = 155, 39.0%), who had estimated scores of 7.31 prior to surgery that increased slightly over 6 months. The third subgroup was named the Moderate level of attentional lapses class (n = 150, 37.8%), who had estimated scores of 5.64 prior to surgery that remained stable over 6 months. The fourth subgroup was named the High level of attentional lapses (n = 22, 5.5%), who had estimated scores of 3.21 that remained stable over 6 months.

#### 3.3.1. Differences in Demographic and Clinical Characteristics

As shown in Table 5, compared to the Very Low level of attentional lapses, the Moderate and High level classes were younger. Compared to the Very Low level of attentional lapses, the High level class was more likely to report a higher comorbidity burden and less likely to have an estrogen receptor positive tumor. Compared to the Very Low level of attentional lapses, the Moderate level class had a higher stage of disease and was more likely to have received neoadjuvant chemotherapy. Compared to the other two attentional lapses classes, the Moderate and High level classes were more likely to have lower functional status. Compared to the Low level of attentional lapses class, the Moderate and High level classes were less likely to have an estrogen receptor positive tumor.

#### 3.3.2. Differences in Psychological and Physical Symptoms

Differences in trait anxiety and state anxiety followed the same pattern (i.e., Very Low and Low < Moderate and High). Differences in depression and decrements in energy followed the same pattern (i.e., Very Low and Low < Moderate < High). Differences in fatigue and sleep disturbance followed the same pattern (i.e., Very Low < Low < Moderate < High). Compared to the Moderate level of attentional lapses class, the High level class reported higher scores for average daily pain, worst pain, and pain interference.

### 3.4. Interpersonal Effectiveness Latent Classes

For interpersonal effectiveness, 59.9% (n = 238) of the patients were in the High interpersonal effectiveness class. These patients had estimated scores of 7.88 and their mean score increased slightly over 6 months. The second subgroup was named Low interpersonal effectiveness class (n = 159, 40.1%). They had estimated scores of 5.25 prior to surgery that remained relatively stable over 6 months. 

#### 3.4.1. Differences in Demographic and Clinical Characteristics 

As shown in Table 6, compared to the High interpersonal effectiveness class, the Low class was younger, less likely to be white, more likely to be Asian or Pacific Islander, more likely to have lower income, and less likely to exercise on a regular basis. Compared to the High interpersonal effectiveness class, the Low class had higher body mass index (BMI), lower functional status, was less likely to have a sentinel node biopsy, more likely to have axillary lymph node dissection, and more likely to be pre-menopausal. 

#### 3.4.2. Differences in Psychological and Physical Symptoms 

Compared to the High interpersonal effectiveness class, the Low class reported higher levels of trait and state anxiety, depression, fatigue, decrements in energy, sleep disturbance, and worst pain intensity and breast pain interference. 

## 4. Discussion

This study is the first to use LCGA to identify subgroups of patients with breast cancer and distinct profiles of CRCI based on cognitive processes that depend on working memory, inhibition control, and cognitive flexibility. CRCI is commonly assessed with objective measures that evaluate working memory, processing speed, attention span, and verbal fluency [2,29]. However, objective neuropsychological tests do not correlate with patients’ self-reports of CRCI [2,29], and the information provided by these tests cannot be easily translated into the specific cognitive deficits experienced by patients with cancer. Our findings provide plausible hypotheses for how scores on the AFI subscales can be equated with deficits in working memory, inhibitory control, and cognitive flexibility. 

Across the three AFI subscales, we evaluated for common and distinct characteristics associated with worse cognitive performance. While for the AFI total score, three classes were identified [26], for the three subscales the number of classes ranged from two (interpersonal effectiveness) to four (attentional lapses). While the exact reasons for the different number of classes cannot be determined, each of our average subscale scores prior to surgery were comparable to those reported in another study [15] (i.e., effective action: 6.23 vs. 6.37; attentional lapses: 6.26 vs. 6.56; and interpersonal effectiveness: 6.57 vs. 6.87). Although clinically meaningful cutoff scores are not available for each of the subscales, if we use the cutoff of <7.5 for the AFI total score to indicate moderate to high levels of cognitive impairment, then 64.2%, 43.3%, and 40.1% of our patients had clinically meaningful decrements in effective action, attentional lapses, and interpersonal effectiveness, respectively. Most importantly, these decrements persisted for 6 months following surgery. The remainder of this discussion focuses on common and distinct characteristics associated with decrements across the cognitive processes assessed using the three subscales (Table 7).

### 4.1. Demographic and Clinical Characteristics 

Lower functional status was the only characteristic that was associated with clinically meaningful decrements in cognitive function across all three of the AFI subscales. These results are consistent with previous reports that found that lower levels of physical functioning were associated with worse attentional function and poorer memory in older adults [47] and in patients with breast cancer [48,49]. Across all three AFI subscales, differences in KPS scores between the highest and the lowest cognitive function classes represented not only statistically significant but clinically meaningful differences (i.e., effective action (d = 0.72), attentional lapses (d = 1.11), interpersonal effectiveness (d = 0.25)) [50]. As noted in one review [51], an individual’s ability to perform physical tasks may be constrained by their level of cognitive function. Our findings support this bidirectional association, between lower functional status and decrements in cognitive processes that depend on working memory, inhibitory control, and cognitive flexibility. Future studies need to examine the mechanisms that underlie these associations. 

It is interesting to note that most of the demographic and clinical characteristics we identified were associated with belonging to the Low interpersonal effectiveness class. The three items on this subscale evaluate how individuals maintain meaningful relationships with others and their responses to lack of inhibitory control. It is not readily apparent why self-identifying as Asian or Pacific Islander was associated with membership in the low interpersonal effectiveness class and warrants evaluation in a more ethnically diverse sample [31]. In our study, a higher BMI and lower levels of physical activity were associated with worse performance on the interpersonal effectiveness subscale, consistent with studies reporting that a higher BMI and lower levels of physical activity were associated with worse performance on measures of information processing and executive functions that require inhibitory control and working memory [52,53,54,55]. Our findings suggest that because these two characteristics are associated with interpersonal effectiveness, they may affect cognitive flexibility as well. Finally, one possible explanation for the association between premenopausal status and belonging to the Low interpersonal effectiveness class is that estrogen protects against neurodegeneration and cognitive deficits [56] and mediates executive processes in the prefrontal cortex [57,58]. Future studies are warranted to determine how increased BMI, decreased physical activity, and estrogen levels affect inhibitory control and cognitive flexibility, which are key components of the interpersonal effectiveness subscale.

### 4.2. Psychological and Physical Symptoms

Pre-surgical levels of anxiety, depression, fatigue, energy deficit, and sleep disturbance that exceeded clinically meaningful cutoffs were common and consistent characteristics for membership in the worst classes across all three of the AFI subscales (i.e., Moderate and Low effective action, Moderate and High level of attentional lapses, and Low interpersonal effectiveness classes). Interestingly, higher levels of fatigue and sleep disturbance that exceeded clinically meaningful cutoffs were associated with all classes across the AFI subscales. These findings are consistent with our previous report that used the AFI total scores [30] and with other studies that found that higher levels of anxiety [12,18,27,59], depression [12,59], fatigue [12,60], and sleep disturbance [12,59,61] were associated with CRCI. These findings suggest that CRCI and common co-occurring symptoms may share the same underlying mechanism(s) that affect working memory, inhibitory control, and cognitive flexibility. 

Consistent with a review that noted that in both preclinical and clinical studies, increased pain was associated with worse executive function [62], the occurrence of presurgical breast pain was primarily associated with the worse effective action classes. These patients had worst pain intensity scores in the moderate range and interference scores in the mild range. The absence of an association between breast pain characteristics and the attentional lapses subscale suggests that pain may be more likely to effect cognitive flexibility. As noted in our previous analysis [63], breast pain prior to surgery was associated with a higher number of breast biopsies in this sample. However, given that evidence exists for associations between breast pain and receipt of neoadjuvant chemotherapy [64,65] and the potential exists that inflammatory mediators may contribute to breast pain, future studies need to evaluate the specific etiologies for breast pain in women prior to breast cancer surgery. Finally, the differential effects of single symptoms and co-occurring symptoms on various cognitive processes warrant further investigation.

### 4.3. Potential Mechanisms Associated with Differences in Cognitive Processes 

Brain imaging studies provide information on changes in brain structure and patterns of neural activities that individuals use when they engage in various cognitive processes that underlie tasks of daily life [66,67,68]. The brain region most associated with executive functions and inhibitory control processes is the prefrontal cortex (PFC) [69,70]. PFC receives information from the hippocampus and other brain regions through large-scale brain networks that include the central executive network (CEN), the salience network (SN), and the default mode network (DMN) [70,71,72]. The CEN is responsible for maintaining and manipulating information in working memory and supports decision making and problem-solving in goal-directed behavior. The SN plays an important role in the detection and selection of salient stimuli, guiding attention and goal-directed behaviors [71,72,73]. The CEN and SN are often co-activated when individuals engage in tasks requiring self-control [74]. The DMN is activated in self-referential processing, and it is deactivated when attention is focused on the external environment and during the formation of working memory [75,76,77]. The DMN is typically deactivated when CEN and SN are activated [74]. This triple network model supports cognitive processes associated with response inhibition, attention, and cognitive flexibility and guides goal-directed behavior [70,71]. 

Population-based studies found that the dysregulation of PFC circuits and hypothalamus hypoactivity are associated with anxiety- and depression-like behaviors and with sleep problems [78,79,80]. Pain influences cognition through activities occurring in the PFC, the hippocampus, and the amygdala; the latter brain region is not involved in other co-occurring symptoms [62]. In studies of patients with non-CNS tumors, including patients with breast cancer, reductions in the volume of grey and white matter in PFC and in parietal and temporal brain regions were associated with compromised performance in working memory and executive function tasks [67,68]. In patients with breast cancer, co-occurring symptoms (e.g., fatigue and depression) were associated with changes in brain connectivity in PFC regions that involve the CEN, SN, and DMN [81,82,83,84]. However, the direction of these associations is not known (i.e., whether symptoms cause changes in brain connectivity, whether changes of brain connectivity cause co-occurring symptoms, or if a bidirectional relationship exists) [84]. 

Another explanation for our findings may be the release of tumor-induced inflammatory cytokines, which lead to altered concentrations of various neurotransmitters (e.g., dopamine and serotonin) [85,86]. Pre-clinical and clinical studies suggest that dopamine, norepinephrine, serotonin, and acetylcholine act directly on the medial PFC and the orbital frontal cortex and regulate various cognitive processes, including attention, inhibitory control, and cognitive flexibility [85,87,88]. Decreased concentrations of dopamine, serotonin, and noradrenaline have been associated with depressive-like behaviors in cancer patients [85]. The tumor microenvironment (TME) induces chronic systemic inflammation, which may disrupt synthesis of neurotransmitters in brain regions responsible for the synthesis of dopamine, serotonin, and noradrenaline [85,87]. Future studies are warranted that examine the associations between TME and alterations in neurotransmitters that may directly or indirectly alter brain function and connectivity, resulting in CRCI. 

### 4.4. Limitations 

Major strengths of the study include its longitudinal design and the use of LCGA to identify subgroups of patients with distinct profiles for each of the AFI subscales. However, study limitations warrant consideration. First, most of the women were well-educated and diagnosed with early-stage breast cancer, which limits the generalizability of our findings. Second, data regarding triple negative breast cancer and chemotherapy regimens were not collected. Given the large amount of inter-individual variability in chemotherapy regimens for breast cancer, any types of meaningful analyses warrant a larger study. Third, the association between self-reported ethnicity and CRCI found in this study needs to be interpreted with caution, given that 65% of our sample self-identified as white. Finally, although LCGA is a powerful statistical procedure, proper class assignment is not guaranteed. Similar LCGA analyses are warranted using objective measures of various cognitive processes.

### 4.5. Implications for Research and Practice

Our study is the first to identify subgroups of patients with distinct trajectories in effective action, attentional lapses, and interpersonal effectiveness assessed using the AFI subscales. Currently, no subjective or objective measures exist to evaluate “real-world” complex tasks (e.g., “I can’t balance my check book”) in oncology patients or survivors. Self-report measures are more likely to detect subtle changes in working memory and inhibitory control at earlier stages of CRCI [30,31,32,33]. Future studies need to determine clinically meaningful cutoff scores for each of the AFI subscales and validate their clinical utility as a screening tool to identify patients who need pretreatment interventions. Measures used in other fields, e.g., an everyday problem-solving inventory that is used in patients with traumatic brain injury [89], warrant evaluation in oncology patients, particularly as measures to evaluate the efficacy of interventions for CRCI.

The use of LCGA allowed us to identify subgroups of patients with worse performance across different cognitive processes. Our findings indicate that classes were consistent over time, meaning that patients with the worst cognitive function did not improve for at least 6 months. Given the consistent associations between higher levels of psychological symptoms and worse cognitive function, clinicians need to perform comprehensive symptom assessments and initiate pharmacological and non-pharmacological interventions (e.g., mindfulness-based stress reduction, cognitive behavioral therapy, exercise, social engagement, sleep hygiene) for patients with these characteristics [2,90]. Moreover, executive function rehabilitation programs, such as the Goal Management Training (GMT), can be implemented at early stages to improve executive functioning, attention, and goal-directive behaviors in patients at increased risk for severe and/or persistent CRCI [91].

## Figures and Tables

**Figure 1 cancers-14-03281-f001:**
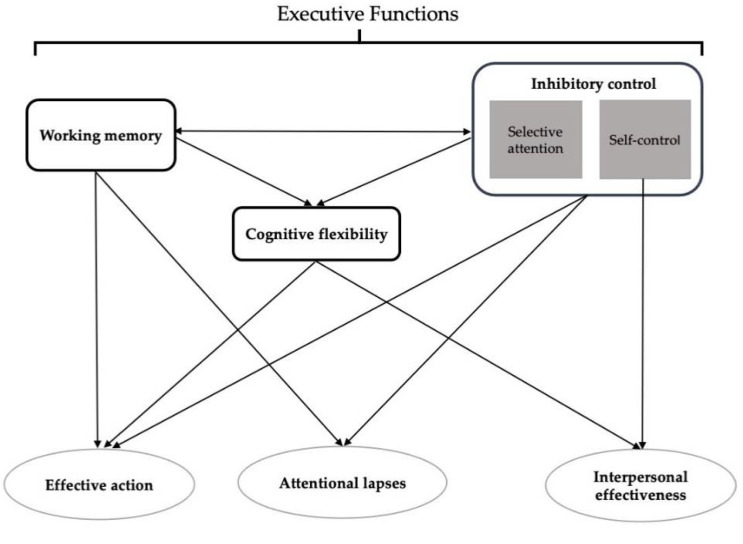
Cognitive processes associated with executive functions (adapted from Diamond [4]).

**Figure 2 cancers-14-03281-f002:**
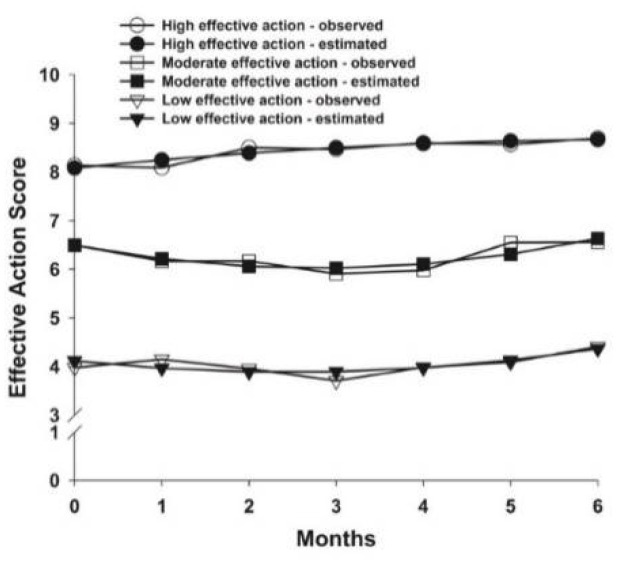
Observed and estimated effective action latent classes from prior to through to 6 months after breast cancer surgery.

**Figure 3 cancers-14-03281-f003:**
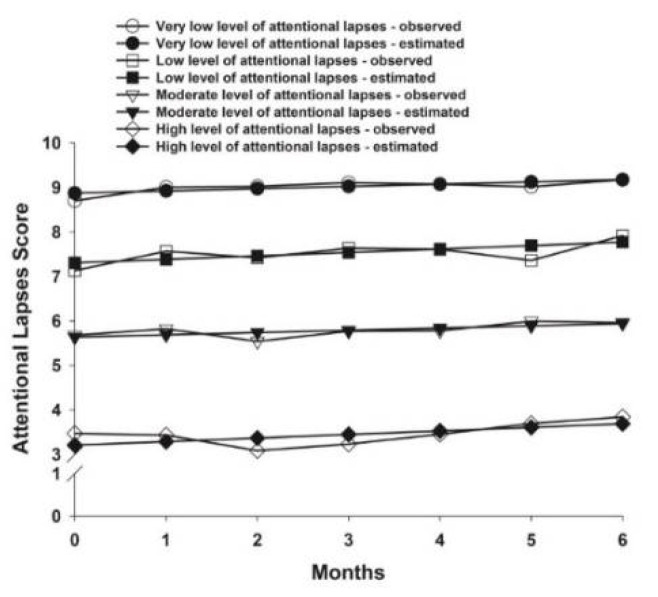
Observed and estimated attentional lapses latent classes from prior to through to 6 months after breast cancer surgery.

**Figure 4 cancers-14-03281-f004:**
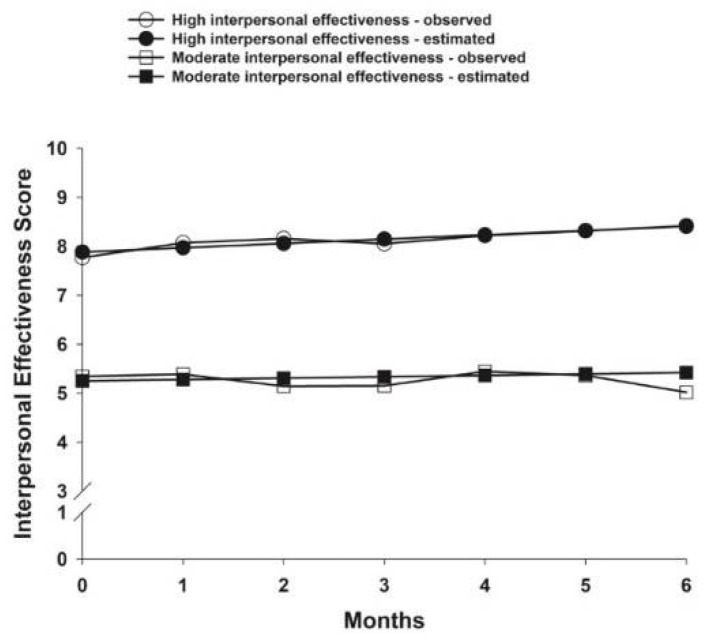
Observed and estimated interpersonal effectiveness latent classes from prior to through to 6 months after breast cancer surgery.

**Table 1 cancers-14-03281-t001:** Associations Between Cognitive Processes and Individual Items for Subscales of Attentional Function Index.

AFI Subscales/Items	Cognitive Processes			
	Working Memory	Selective Attention	Self-Control	Cognitive Flexibility
Effective action subscale (purposeful actions: reasoning, planning, executing, problem-solving)				
Getting started on activities (tasks, jobs) you intend to do	♦		♦	
Following through on your plans	♦	♦	♦	♦
Doing things that take time and effort	♦	♦	♦	♦
Making mind up about things	♦			♦
Keeping your mind on what you are doing	♦	♦	♦	
Remembering to do all the things you started out to do	♦	♦	♦	
Keeping your mind on what others are saying	♦	♦	♦	
Attentional lapses subscale (difficulties in inhibiting distraction, i.e., selective attention)				
How hard do you find it to concentrate on details		♦		
How often do you make mistakes on what you are doing	♦	♦	♦	
How often do you forget to do important things	♦	♦		
Interpersonal effectiveness (maintaining interpersonal relationships & responding to lack of inhibitory control)				
Keeping yourself from saying or doing things you did not want to say or do			♦	
Being patient with others			♦	♦
How often do you get easily annoyed or irritated			♦	♦

**Table 2 cancers-14-03281-t002:** Fit Indices for Effective Action, Attentional Lapses, and Interpersonal Effectiveness Subscales of the Attentional Function Index Based on LCGA solutions for Seven Assessments of Patients with Breast Cancer From Prior to Through to 6 Months After Surgery.

LCGM	LL	AIC	BIC	Entropy	VLMR
Effective Action					
1-class ^a^	−5574.88	11,169.77	11,209.61	n/a	n/a
2-class	−5043.39	10,114.78	10,170.55	0.85	1062.99 ^‡^
3-class ^b^	−4867.38	9770.76	9842.47	0.86	352.02 ^‡^
4-class	−4836.81	9717.61	9805.26	0.79	ns
Attentional Lapses					
1-class ^a^	−5377.95	10,773.89	10,809.75	n/a	n/a
2-class	−5001.77	10,027.53	10,075.34	0.83	752.36 ^‡^
3-class	−4901.48	9832.95	9892.71	0.85	200.58 ^‡^
4-class ^c^	−4857.80	9751.19	9822.91	0.80	87.76 ***
5-class	−4846.08	9734.17	9817.83	0.82	n/a
Interpersonal Effectiveness					
1-class ^a^	−5348.14	10,714.28	10,750.13	n/a	n/a
2-class ^d^	−4854.08	9732.15	9779.96	0.88	988.13 ^‡^
3-class	−4740.19	9510.39	9570.15	0.82	ns

*** *p* < 0.001; ^‡^
*p* < 0.00005. ^a^ Baseline linear growth curve. Entropy, BLRT, and VLMR are not estimated. ^b^ For the affective action subscale a 3-class model was selected. The BIC for the 3-class model was smaller than for the 2-class solution and the VLMR for the 3-class solution was significant, both indicating a better fit to the data for the 3-class solution. Although the BIC for the 4-class solution was smaller than for the 3-class solution, the VLMR for the 4-class solution was not significant, indicating that too many classes were extracted. ^c^ For the attentional lapses subscale a 4-class model was selected. The BIC for the 4-class model was smaller than for the 3-class model indicating a better fit to the data. The VLMR indicates that four classes fit the data better than three classes. The 5-class model produced a class with only two cases and was rejected as unlikely to generalize to other samples. ^d^ For the interpersonal effectiveness subscale a 2-class model was selected. The BIC for the 2-class model was smaller than for the 1-class (baseline) solution and the VLMR for the 2-class solution was significant, both indicating a better fit to the data for the 2-class solution. Although the BIC for the 3-class solution was smaller than for the 2-class solution, the VLMR for the 3-class solution was not significant, indicating that too many classes were extracted. Abbreviations: AIC = Akaike Information Criterion; BIC = Bayesian Information Criterion; LCGA = latent class growth analysis; LCGM = latent class growth model; LL = loglikelihood; n/a = not applicable; ns = not significant; VLMR = Vuong-Lo-Mendell-Rubin likelihood ratio test.

**Table 3 cancers-14-03281-t003:** LCGA Parameter Estimates for the Latent Class Solutions for the Effective Action, Attentional Lapses, and Interpersonal Effectiveness Subscale Scores.

Effective Action ^a^				
Parameter Estimates ^b^	High n = 142 (35.8%)	Moderate n = 160 (40.3%)		Low n = 95 (23.9%)
	Mean (SE)	Mean (SE)		(Mean SE)
Intercept	8.08 (0.14) ^‡^	6.50 (0.16) ^‡^		4.12 (0.26) ^‡^
Linear slope	0.02 (0.08) *	−0.34 (0.12) **		−0.19 (0.15)
Quadratic slope	−0.01 (0.01)	0.06 (0.02) ***		0.04 (0.03)
Attentional Lapses ^a^				
Parameter Estimates ^b^	Very Low Level n = 70 (17.6%)	Low Level n = 155 (39.0%)	Moderate Level n = 150 (37.8%)	High Level n = 22 (5.5%)
	Mean (SE)	Mean (SE)	Mean (SE)	Mean (SE)
Intercept	8.87 (0.14) ***	7.31 (0.12) ***	5.64 (0.14) ***	3.21 (0.45) ***
Linear slope	0.05 (0.02) *	0.08 (0.03) *	0.05 (0.04)	0.08 (0.09)
Interpersonal Effectiveness ^a^				
Parameter Estimates ^b^	High n = 238 (59.9%)		Moderate n = 159 (40.1%)	
	Mean (SE)		Mean (SE)	
Intercept	7.88 (0.10) ^‡^		5.25 (0.17) ^‡^	
Linear slope	0.09 (0.02) ^‡^		0.03 (0.03)	

* *p* < 0.05, ** *p* < 0.01, *** *p* ≤ 0.001, ^‡^ *p* < 0.005. ^a^ Predicted class sizes are based on their most likely class membership. ^b^ Variance and covariance parameter estimates were fixed at zero. Abbreviations: LCGA = latent class growth analysis; SE = standard error.

**Table 4 cancers-14-03281-t004:** Differences in Demographic, Clinical, and Symptom Characteristics among the Effective Action Classes.

Characteristic	High Effective Action (0) n = 142 (35.8%)	Moderate Effective Action (1) n = 160 (40.3%)	Low Effective Action (2) n = 95 (23.9%)	Statistics
	Mean (SD)	Mean (SD)	Mean (SD)	
Demographic and Clinical Characteristics				
Age (years)	57.2 (11.3)	53.3 (11.2)	54.5 (12.1)	F = 4.44, *p* = 0.012 0 > 1
Education (years)	15.7 (2.7)	15.9 (2.8)	15.5 (2.3)	F = 0.57, *p* = 0.566
Self-Administered Comorbidity Questionnaire score	3.8 (2.5)	4.1 (2.4)	5.4 (3.6)	F = 10.20, *p* < 0.001 0 and 1 < 2
Body mass index (kilograms/meter squared)	26.1 (5.8)	27.1 (6.4)	27.4 (6.4)	F = 1.74, *p* = 0.178
Karnofsky Performance Status score	96.2 (7.9)	93.4 (9.6)	88.6 (12.7)	F = 16.43, *p* < 0.001 0 and 1 > 2
	n (%)	n (%)	n (%)	
Race/ethnicity				χ^2^ = 10.81, *p* = 0.094
White	97 (68.8)	109 (68.6)	49 (51.6)	
Black	13 (9.2)	12 (7.5)	15 (15.8)	
Asian/Pacific Islander	17 (12.1)	19 (11.9)	14 (14.7)	
Hispanic/mixed/other	14 (9.9)	19 (11.9)	17 (17.9)	
Live alone (% yes)	25 (18.0)	41 (25.6)	28 (30.1)	χ^2^ = 4.89, *p* = 0.087
Married or partnered (% yes)	52 (36.9)	69 (43.1)	44 (47.3)	χ^2^ = 2.68, *p* = 0.262
Currently employed (% yes)	72 (51.4)	83 (51.9)	34 (36.2)	χ^2^ = 6.89, *p* = 0.032 1 > 2
Household income level				KW = 7.64, *p* = 0.022 0 > 2
<$30,000	17 (14.9)	25 (18.1)	28 (36.4)	
$30,000–$99,999	51 (44.7)	58 (42.0)	25 (32.5)	
≥$100,000	46 (40.4)	55 (39.9)	24 (31.2)	
Regular exercise (% yes)	100 (70.4)	114 (72.2)	60 (63.8)	χ^2^ = 2.01, *p* = 0.366
Occurrence of comorbid conditions				
Heart disease	8 (5.6)	2 (1.3)	5 (5.3)	χ^2^ = 4.73, *p* = 0.094
High blood pressure	51 (35.9)	40 (25.0)	32 (33.7)	χ^2^ = 4.62, *p* = 0.099
Lung disease	4 (2.8)	5 (3.1)	3 (3.2)	χ^2^ = 0.03, *p* = 0.984
Diabetes	9 (6.3)	9 (5.6)	13 (13.7)	χ^2^ = 6.04, *p* = 0.049 No significant pairwise contrasts
Ulcer	3 (2.1)	6 (3.8)	6 (6.3)	χ^2^ = 2.77, *p* = 0.251
Kidney disease	2 (1.4)	0 (0.0)	1 (1.1)	χ^2^ = 2.14, *p* = 0.344
Liver disease	4 (2.8)	4 (2.5)	2 (2.1)	χ^2^ = 0.12, *p* = 0.943
Anemia	10 (7.0)	13 (8.1)	8 (8.4)	χ^2^ = 0.19, *p* = 0.910
Depression	21 (14.8)	35 (21.9)	30 (31.6)	χ^2^ = 9.46, *p* = 0.009 0 < 2
Osteoarthritis	22 (15.5)	24 (15.0)	23 (24.2)	χ^2^ = 4.07, *p* = 0.131
Back pain	33 (23.2)	43 (26.9)	35 (36.8)	χ^2^ = 5.39, *p* = 0.068
Rheumatoid arthritis	2 (1.4)	5 (3.1)	7 (7.4)	χ^2^ = 6.07, *p* = 0.048 No significant pairwise contrasts
Postmenopausal (% yes)	91 (65.5)	98 (63.2)	59 (64.1)	χ^2^ = 0.16, *p* = 0.923
Stage of disease				KW = 11.07, *p* = 0.004 0 < 1
Stage 0	34 (23.9)	22 (13.8)	17 (17.9)	
Stage I	61 (43.0)	57 (35.6)	33 (34.7)	
Stage II	39 (27.5)	65 (40.6)	37 (38.9)	
Stage III and IV	8 (5.6)	16 (10.0)	8 (8.4)	
Receipt of neoadjuvant therapy (% yes)	22 (15.6)	34 (21.3)	23 (24.2)	χ^2^ = 2.92, *p* = 0.233
HRT prior to surgery (% yes)	18 (12.7)	35 (22.0)	14 (14.9)	χ^2^ = 5.02, *p* = 0.081
Type of surgery				χ^2^ = 0.11, *p* = 0.947
Breast conservation	114 (80.3)	127 (79.4)	77 (81.1)	
Mastectomy	28 (19.7)	33 (20.6)	18 (18.9)	
Sentinel node biopsy (% yes)	118 (83.1)	136 (85.0)	74 (77.9)	χ^2^ = 2.13, *p* = 0.345
Axillary lymph node dissection (% yes)	43 (30.5)	65 (40.6)	40 (42.1)	χ^2^ = 4.48, *p* = 0.106
Receipt of adjuvant chemotherapy (% yes) ^a^	34 (23.9)	67 (41.9)	32 (33.7)	χ^2^ = 10.86, *p* = 0.004 0 < 1
Receipt of radiation therapy (% yes) ^a^	81 (57.0)	90 (56.3)	53 (55.8)	χ^2^ = 0.04, *p* = 0.980
Receipt of hormonal therapy (% yes)	65 (45.8)	72 (45.0)	31 (32.6)	χ^2^ = 4.82, *p* = 0.090
Estrogen receptor positive (% positive)	116 (82.3)	128 (80.0)	63 (66.3)	χ^2^ = 9.24, *p* = 0.010 0 and 1 > 2
Progesterone receptor positive (% positive)	108 (76.6)	115 (71.9)	56 (58.9)	χ^2^ = 8.75, *p* = 0.013 0 > 2
HER2/neu (% positive)	16 (12.9)	29 (20.1)	14 (15.6)	χ^2^ = 2.61, *p* = 0.271
Psychological Symptoms *				
	Mean (SD)	Mean (SD)	Mean (SD)	
Trait anxiety (≥31.8)	31.4 (7.1)	35.1 (8.0)	41.7 (9.8)	F = 42.96, *p* < 0.001 0 < 1 < 2
State anxiety (≥32.2)	36.9 (12.8)	41.4 (12.7)	49.5 (12.0)	F = 27.60, *p* < 0.001 0 < 1 < 2
Center for Epidemiological Studies- Depression (≥16.0)	9.4 (7.5)	13.1 (8.5)	21.1 (9.9)	F = 51.95, *p* < 0.001 0 < 1 < 2
Physical Symptoms *				
	Mean (SD)	Mean (SD)	Mean (SD)	
Lee Fatigue Scale-Fatigue (≥4.4)	1.9 (1.9)	3.3 (2.1)	4.6 (2.4)	F = 46.70, *p* < 0.001 0 < 1 < 2
Lee Energy Scale-Energy (≤4.8)	6.1 (2.6)	4.7 (2.0)	3.5 (2.1)	F = 36.13, *p* < 0.001 0 > 1 > 2
General Sleep Disturbance Scale (≥43.0)	36.3 (18.4)	50.3 (18.8)	62.6 (19.5)	F = 55.51, *p* < 0.001 0 < 1 < 2
Pain	n (%)	n (%)	n (%)	
Occurrence of pain in the affected breast prior to surgery (% yes)	25 (18.1)	57 (35.8)	27 (29.3)	χ^2^ = 11.62, *p* = 0.003 0 < 1
For patients with breast pain	Mean (SD)	Mean (SD)	Mean (SD)	
Pain right now	1.2 (1.6)	1.4 (1.5)	2.5 (2.8)	F = 3.89, *p* = 0.024 1 < 2
Current average daily pain	1.8 (1.5)	1.7 (1.5)	3.5 (2.7)	F = 8.34, *p* < 0.001 0 and 1 < 2
Worst pain	3.1 (1.9)	2.9 (1.9)	5.1 (2.7)	F = 9.07, *p* < 0.001 0 and 1 < 2
Number of days per week in pain	2.1 (2.4)	2.4 (2.7)	4.3 (2.5)	F = 6.03, *p* = 0.003 0 and 1 < 2
Breast Pain Interference	0.9 (2.0)	1.2 (1.7)	3.1 (2.4)	F = 10.20, *p* < 0.001 0 and 1 < 2

^a^ Receipt of treatment in the six months following surgery. * Numbers in parentheses indicate clinically meaningful cutoff scores. Abbreviations: HER2/neu = human epidermal growth factor receptor, HRT = hormone replacement therapy, KW = Kruskal–Wallis test, SD = standard deviation

**Table 5 cancers-14-03281-t005:** Differences in Demographic, Clinical, and Symptom Characteristics Among the Attentional Lapses Classes.

Characteristic	Very Low Level of Attentional Lapses (0) n = 70 (17.6%)	Low Level of Attentional Lapses (1) n = 155 (39.0%)	Moderate Level of Attentional Lapses (2) n = 150 (37.8%)	High Level of Attentional Lapses (3) n = 22 (5.5%)	Statistics
Demographic and Clinical Characteristics					
	Mean (SD)	Mean (SD)	Mean (SD)	Mean (SD)	
Age (years)	60.5 (10.9)	54.6 (10.7)	53.8 (12.1)	48.0 (10.0)	F = 9.14, *p* < 0.001 0 > 2 and 3
Education (years)	15.5 (2.6)	15.9 (2.7)	15.5 (2.7)	16.5 (2.1)	F = 1.26, *p* = 0.288
Self-Administered Comorbidity Questionnaire score	3.8 (2.6)	4.0 (2.5)	4.6 (2.9)	5.6 (4.0)	F = 3.78, *p* = 0.011 0 < 3
Body mass index (kilograms/meter squared)	26.0 (4.8)	26.7 (6.7)	27.2 (6.1)	27.2 (6.9)	F = 0.70, *p* = 0.552
Karnofsky Performance Status score	97.7 (5.5)	94.9 (8.8)	91.0 (10.7)	83.6 (17.1)	F = 16.04, *p* < 0.001 0 and 1 > 2 > 3
	n (%)	n (%)	n (%)	n (%)	
Race/ethnicity					χ^2^ = 14.30, *p* = 0.112
White	53 (75.7)	101 (66.0)	84 (56.0)	17 (77.3)	
Black	7 (10.0)	11 (7.2)	21 (14.0)	1 (4.5)	
Asian/Pacific Islander	4 (5.7)	22 (14.4)	23 (15.3)	1 (4.5)	
Hispanic/mixed/other	6 (8.6)	19 (12.4)	22 (14.7)	3 (13.6)	
Live alone (% yes)	19 (28.4)	32 (20.8)	34 (22.8)	9 (40.9)	χ^2^ = 5.14, *p* = 0.162
Married or partnered (% yes)	28 (40.6)	61 (39.6)	64 (43.0)	12 (54.5)	χ^2^ = 1.89, *p* = 0.595
Currently employed (% yes)	35 (50.0)	78 (51.0)	67 (45.0)	9 (40.9)	χ^2^ = 1.65, *p* = 0.648
Household income level					KW = 4.57, *p* = 0.206
<$30,000	6 (10.9)	24 (18.2)	33 (26.4)	7 (41.2)	
$30,000–$99,999	26 (47.3)	56 (42.4)	48 (38.4)	4 (23.5)	
≥$100,000	23 (41.8)	52 (39.4)	44 (35.2)	6 (35.3)	
Regular exercise (% yes)	49 (70.0)	110 (71.4)	103 (69.6)	12 (54.5)	χ^2^ = 2.60, *p* = 0.457
Occurrence of comorbid conditions					
Heart disease	4 (5.7)	6 (3.9)	4 (2.7)	1 (4.5)	χ^2^ = 1.27, *p* = 0.736
High blood pressure	30 (42.9)	44 (28.4)	46 (30.7)	3 (13.6)	χ^2^ = 8.21, *p* = 0.042 No significant pairwise contrasts
Lung disease	2 (2.9)	4 (2.6)	4 (2.7)	2 (9.1)	χ^2^ = 2.94, *p* = 0.401
Diabetes	1 (1.4)	14 (9.0)	15 (10.0)	1 (4.5)	χ^2^ = 5.61, *p* = 0.132
Ulcer	2 (2.9)	4 (2.6)	7 (4.7)	2 (9.1)	χ^2^ = 2.81, *p* = 0.422
Kidney disease	2 (2.9)	1 (0.6)	0 (0.0)	0 (0.0)	χ^2^ = 5.46, *p* = 0.141
Liver disease	0 (0.0)	4 (2.6)	4 (2.7)	2 (9.1)	χ^2^ = 5.69, *p* = 0.127
Anemia	4 (5.7)	10 (6.5)	13 (8.7)	4 (18.2)	χ^2^ = 4.27, *p* = 0.234
Depression	9 (12.9)	33 (21.3)	37 (24.7)	7 (31.8)	χ^2^ = 5.35, *p* = 0.148
Osteoarthritis	13 (18.6)	23 (14.8)	28 (18.7)	5 (22.7)	χ^2^ = 1.38, *p* = 0.711
Back pain	15 (21.4)	43 (27.7)	43 (28.7)	10 (45.5)	χ^2^ = 4.87, *p* = 0.182
Rheumatoid arthritis	1 (1.4)	5 (3.2)	5 (3.3)	3 (13.6)	χ^2^ = 7.57, *p* = 0.056
Postmenopausal (% yes)	50 (73.5)	91 (60.7)	93 (63.7)	14 (63.6)	χ^2^ = 3.41, *p* = 0.333
Stage of disease					KW = 12.95, *p* = 0.005 0 < 2
Stage 0	18 (25.7)	29 (18.7)	22 (14.7)	4 (18.2)	
Stage I	34 (48.6)	59 (38.1)	51 (34.0)	7 (31.8)	
Stage II	15 (21.4)	57 (36.8)	60 (40.0)	9 (40.9)	
Stage III and IV	3 (4.3)	10 (6.5)	17 (11.3)	2 (9.1)	
Receipt of neoadjuvant therapy (% yes)	7 (10.0)	27 (17.5)	40 (26.7)	5 (22.7)	χ^2^ = 9.25, *p* = 0.026 0 < 2
HRT prior to surgery (% yes)	10 (14.3)	26 (16.9)	24 (16.0)	7 (33.3)	χ^2^ = 4.45, *p* = 0.217
Type of surgery					χ^2^ = 2.53, *p* = 0.470
Breast conservation	60 (85.7)	122 (78.7)	117 (78.0)	19 (86.4)	
Mastectomy	10 (14.3)	33 (21.3)	33 (22.0)	3 (13.6)	
Sentinel node biopsy (% yes)	61 (87.1)	130 (83.9)	121 (80.7)	16 (72.7)	χ^2^ = 3.06, *p* = 0.382
Axillary lymph node dissection (% yes)	18 (25.7)	58 (37.7)	61 (40.7)	11 (50.0)	χ^2^ = 6.27, *p* = 0.099
Receipt of adjuvant chemotherapy (% yes) ^a^	13 (18.6)	56 (36.1)	52 (34.7)	12 (54.5)	χ^2^ = 11.95, *p* = 0.008 0 < 1 and 3
Receipt of radiation therapy (% yes) ^a^	44 (62.9)	80 (51.6)	86 (57.3)	14 (63.6)	χ^2^ = 3.15, *p* = 0.369
Receipt of hormonal therapy (% yes)	34 (48.6)	68 (43.9)	59 (39.3)	7 (31.8)	χ^2^ = 2.82, *p* = 0.421
Estrogen receptor positive (% positive)	59 (84.3)	130 (84.4)	107 (71.3)	11 (50.0)	χ^2^ = 18.90, *p* < 0.001 0 > 3; 1 > 2 and 3
Progesterone receptor positive (% positive)	55 (78.6)	115 (74.7)	98 (65.3)	11 (50.0)	χ^2^ = 9.85, *p* = 0.020 No significant pairwise contrasts
HER2/neu (% positive)	5 (8.6)	28 (20.0)	20 (14.3)	6 (30.0)	χ^2^ = 7.01, *p* = 0.072
Psychological Symptoms *					
	Mean (SD)	Mean (SD)	Mean (SD)	Mean (SD)	
Trait anxiety (≥31.8)	30.8 (8.0)	33.7 (7.7)	38.3 (9.0)	40.9 (11.4)	F = 16.91, *p* < 0.001 0 and 1 < 2 and 3
State anxiety (≥32.2)	35.9 (13.9)	39.8 (12.5)	45.3 (12.7)	49.3 (12.9)	F = 11.94, *p* < 0.001 0 and 1 < 2 and 3
Center for Epidemiological Studies- Depression (≥16.0)	8.4 (8.0)	11.4 (7.7)	17.0 (9.6)	23.1 (11.5)	F = 26.85, *p* < 0.001 0 and 1 < 2 < 3
Physical Symptoms *					
	Mean (SD)	Mean (SD)	Mean (SD)	Mean (SD)	
Lee Fatigue Scale-Fatigue (≥4.4)	1.6 (1.8)	2.5 (2.0)	4.0 (2.1)	6.3 (2.3)	F = 44.17, *p* < 0.001 0 < 1 < 2 < 3
Lee Energy Scale-Energy (≤4.8)	6.1 (3.0)	5.2 (2.3)	4.4 (2.0)	2.8 (2.1)	F = 14.46, *p* < 0.001 0 and 1 > 2 > 3
General Sleep Disturbance Scale (≥43.0)	34.5 (18.5)	44.3 (19.1)	55.3 (19.9)	68.9 (20.5)	F = 28.34, *p* < 0.001 0 < 1 < 2 < 3
Pain	n (%)	n (%)	n (%)	n (%)	
Occurrence of pain in the affected breast prior to surgery (% yes)	12 (17.9)	44 (28.9)	50 (33.8)	3 (13.6)	χ^2^ = 8.15, *p* = 0.043 No significant pairwise contrasts
For patients with breast pain	Mean (SD)	Mean (SD)	Mean (SD)	Mean (SD)	
Pain right now	1.8 (3.0)	1.1 (1.3)	2.0 (2.1)	0.7 (1.2)	F = 1.92, *p* = 0.132
Current average daily pain	2.8 (2.7)	1.4 (1.3)	2.7 (2.2)	1.7 (1.2)	F = 3.79, *p* = 0.013 1 < 2
Worst pain	3.5 (2.5)	2.6 (1.5)	4.3 (2.6)	3.7 (1.5)	F = 4.53, *p* = 0.005 1 < 2
Number of days per week in pain	1.6 (2.8)	2.3 (2.7)	3.6 (2.6)	2.0 (1.0)	F = 2.78, *p* = 0.045 No significant pairwise contrasts
Breast Pain Interference	0.5 (1.0)	0.9 (1.5)	1.9 (2.1)	2.4 (1.4)	F = 3.94, *p* = 0.010 1 < 2

^a^ Receipt of treatment in the six months following surgery. * Numbers in parentheses indicate clinically meaningful cutoff scores. Abbreviations: HER2/neu = human epidermal growth factor receptor, HRT = hormone replacement therapy, KW = Kruskal–Wallis test, SD = standard deviation.

**Table 6 cancers-14-03281-t006:** Differences in Demographic, Clinical, and Symptom Characteristics Among the Interpersonal Effectiveness Classes.

Characteristic	High Interpersonal Effectiveness (0) n = 238 (59.9%)	Low Interpersonal Effectiveness (1) n = 159 (40.1%)	Statistics
Demographic and Clinical Characteristics			
	Mean (SD)	Mean (SD)	
Age (years)	56.5 (10.9)	52.7 (12.2)	t = 3.27, *p* = 0.001 0 > 1
Education (years)	15.8 (2.6)	15.6 (2.7)	t = 0.53, *p* = 0.596
Self-Administered Comorbidity Questionnaire score	4.1 (2.7)	4.6 (3.0)	t=−1.73, *p* = 0.085
Body mass index (kilograms/meter squared)	26.1 (5.3)	27.9 (7.1)	t = −2.72, *p* = 0.007 0 < 1
Karnofsky Performance Status score	94.3 (9.6)	91.7 (11.2)	t = 2.48, *p* = 0.014 0 > 1
	n (%)	n (%)	
Race/ethnicity			(2 = 18.20, *p* < 0.001
White	172 (72.9)	83 (52.2)	0 > 1
Black	19 (8.1)	21 (13.2)	NS
Asian/Pacific Islander	21 (8.9)	29 (18.2)	0 < 1
Hispanic/mixed/other	24 (10.2)	26 (16.4)	NS
Live alone (% yes)	57 (24.4)	37 (23.4)	FE, *p* = 0.904
Married or partnered (% yes)	95 (40.3)	70 (44.3)	FE, *p* = 0.466
Currently employed (% yes)	119 (50.4)	70 (44.3)	FE, *p* = 0.258
Household income level			U, *p* = 0.003
<$30,000	33 (16.6)	37 (28.5)	
$30,000–$99,999	80 (40.2)	54 (41.5)	
≥$100,000	86 (43.2)	39 (30.0)	
Regular exercise (% yes)	174 (73.4)	100 (63.7)	FE, *p* = 0.045 0 > 1
Occurrence of comorbid conditions			
Heart disease	10 (4.2)	5 (3.1)	FE, *p* = 0.789
High blood pressure	75 (31.5)	48 (30.2)	FE, *p* = 0.825
Lung disease	7 (2.9)	5 (3.1)	FE, *p* = 1.000
Diabetes	18 (7.6)	13 (8.2)	FE, *p* = 0.850
Ulcer	7 (2.9)	8 (5.0)	FE, *p* = 0.296
Kidney disease	2 (0.8)	1 (0.6)	FE, *p* = 1.000
Liver disease	7 (2.9)	3 (1.9)	FE, *p* = 0.746
Anemia	18 (7.6)	13 (8.2)	FE, *p* = 0.850
Depression	44 (18.5)	42 (26.4)	FE, *p* = 0.063
Osteoarthritis	41 (17.2)	28 (17.6)	FE, *p* =1.000
Back pain	62 (26.1)	49 (30.8)	FE, *p* = 0.307
Rheumatoid arthritis	6 (2.5)	8 (5.0)	FE, *p* = 0.266
Postmenopausal (% yes)	164 (70.4)	84 (54.9)	FE, *p* = 0.002 0 > 1
Stage of disease			U, *p* = 0.227
Stage 0	45 (18.9)	28 (17.6)	
Stage I	97 (40.8)	54 (34.0)	
Stage II	77 (32.4)	64 (40.3)	
Stage III and IV	19 (8.0)	13 (8.2)	
Receipt of neoadjuvant therapy (% yes)	42 (17.7)	37 (23.3)	FE, *p* = 0.200
HRT prior to surgery (% yes)	41 (17.3)	26 (16.5)	FE, *p* = 0.892
Type of surgery			FE, *p* = 0.608
Breast conservation	193 (81.1)	125 (78.6)	
Mastectomy	45 (18.9)	34 (21.4)	
Sentinel node biopsy (% yes)	206 (86.6)	122 (76.7)	FE, *p* = 0.015 0 > 1
Axillary lymph node dissection (% yes)	79 (33.3)	69 (43.4)	FE, *p* = 0.045 0 < 1
Receipt of adjuvant chemotherapy (% yes) ^a^	72 (30.3)	61 (38.4)	FE, *p* = 0.104
Receipt of radiation therapy (% yes) ^a^	137 (57.6)	87 (54.7)	FE, *p* = 0.606
Receipt of hormonal therapy (% yes)	107 (45.0)	61 (38.4)	FE, *p* = 0.214
Estrogen receptor positive (% positive)	188 (79.3)	119 (74.8)	FE, *p* = 0.326
Progesterone receptor positive (% positive)	172 (72.6)	107 (67.3)	FE, *p* = 0.264
HER2/neu (% positive)	35 (16.5)	24 (16.4)	FE, *p* = 1.000
Psychological Symptoms *			
	Mean (SD)	Mean (SD)	
Trait anxiety (≥31.8)	32.8 (8.1)	39.1 (9.0)	t = 7.01, *p* < 0.001 0 < 1
State anxiety (≥32.2)	38.9 (13.0)	45.8 (13.1)	t = −5.05, *p* < 0.001 0 < 1
Center for Epidemiological Studies—Depression (≥16.0)	11.1 (8.7)	17.4 (9.6)	t = −6.59, *p* < 0.001 0 < 1
Physical Symptoms *			
	Mean (SD)	Mean (SD)	
Lee Fatigue Scale-Fatigue (≥4.4)	2.6 (2.2)	3.9 (2.4)	t = −5.68, *p* < 0.001 0 < 1
Lee Energy Scale-Energy (≤4.8)	5.3 (2.6)	4.4 (2.1)	t = 3.84, *p* < 0.001 0 > 1
General Sleep Disturbance Scale (≥43.0)	42.6 (20.3)	56.4 (20.4)	t = −6.48, *p* < 0.001 0 < 1
Pain	n (%)	n (%)	
Occurrence of pain in the affected breast prior to surgery (% yes)	62 (26.6)	47 (30.1)	FE, *p* = 0.490
For patients with breast pain	Mean (SD)	Mean (SD)	
Pain right now	1.3 (1.9)	1.9 (2.0)	t = −1.52, *p* = 0.131
Current average daily pain	1.9 (1.9)	2.5 (2.1)	t = −1.62, *p* = 0.109
Worst pain	3.1 (1.9)	4.1 (2.6)	t = −2.17, *p* = 0.033 0 < 1
Number of days per week in pain	2.4 (2.7)	3.4 (2.7)	t = −1.83, *p* = 0.070
Breast Pain Interference	1.2 (1.8)	2.2 (2.4)	t = −2.44, *p* = 0.017 0 < 1

^a^ Receipt of treatment in the six months following surgery. * Numbers in parentheses indicate clinically meaningful cutoff scores. Abbreviations: FE = Fisher’s Exact test, HER2/neu = human epidermal growth factor receptor, HRT = hormone replacement therapy, NS = not significant, SD = standard deviation.

**Table 7 cancers-14-03281-t007:** Characteristics Associated with Membership Among the Effective Action, Attentional Lapses, and Interpersonal Effectiveness Classes.

Characteristic ^a^	Effective Action		Attentional Lapses			Interpersonal Effectiveness
	Moderate Effective Action	Low Effective Action	Low Level of Attentional Lapses	Moderate Level of Attentional Lapses	High level of Attentional Lapses	Low Interpersonal Effectiveness
Demographic characteristics						
Younger age	♦			♦	♦	♦
Less likely to be white						♦
More likely to be Asian/Pacific Islander						♦
More likely to have a lower annual income		♦				♦
Less likely to exercise on a regular basis						♦
Clinical characteristics						
Higher body mass index						♦
Higher comorbidity burden		♦			♦	
Lower functional status		♦		♦	♦	♦
More likely to self-report depression		♦				
More likely to be diagnosed with higher stage disease	♦			♦		
Less likely to undergone menopause						♦
Less likely to have had sentinel node biopsy						♦
More likely to have had axillary lymph node dissection						♦
More likely to have received neoadjuvant therapy				♦		
More likely to have received adjuvant chemotherapy in the 6 months after surgery	♦		♦		♦	
Less likely to be positive in estrogen receptor		♦			♦	
Less likely to be positive in progesterone receptor		♦				
Psychological symptoms						
Higher trait anxiety	♦	♦		♦	♦	♦
Higher state anxiety	♦	♦		♦	♦	♦
Higher depression symptoms	♦	♦		♦	♦	♦
Physical symptoms						
Higher fatigue	♦	♦	♦	♦	♦	♦
Lower energy	♦	♦		♦	♦	♦
Higher sleep disturbance	♦	♦	♦	♦	♦	♦
More likely to have pain in the affected breast prior to surgery	♦					
Higher average daily pain		♦				
Higher worst pain intensity		♦				♦
Higher number of days per week in pain		♦				
Higher pain interference		♦				♦

^a^ Comparisons done with the High Effective Action, Very Low Level of Attentional Lapses, and High Interpersonal Effectiveness Groups. ♦ Comparisons were significant.

## Data Availability

Data availability is available on request and following the completion of a material transfer agreement with Miaskowski.

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
