# Peer review of "Pre-Surgery Demographic, Clinical, and Symptom Characteristics Associated with Different Self-Reported Cognitive Processes in Patients with Breast Cancer"

_cancers, 2022, doi:10.3390/cancers14133281_

Round 1

Reviewer 1 Report

The authors report the results of an interesting but complex study.

They give many important details  concerning the characteristics of breast tumors. They mention  the percentage of estrogen and or progesterone receptor positive breast cancer and HER2 positive breast cancer. I am sure they have the information but it is very important to add  data about the percentage of triple negative breast cancer (table 4)

- In the same context, chemotherapy regimen need to be detailed and the number of courses received

In the section materials and methods the authors mention  that the baseline evaluation was performed +/- 4 days before surgery. If I understand correctly , in table 4, they mention that a non neglectable percentage of patients (15 to 24% of patients according the group) received neoadjuvant chemotherapy and had a baseline assesment  after receiving neoadjuvant chemotherapy. If there is no initial evaluation -before administration of any treatment, I think these patients need to be excluded because it is not a baseline evaluation

Whatever the mode of administration of the chemotherapy, we know this  treatment has an important impact on anxiety, cognitive function, depression, pain ...The best option would have been to perform the baseline evaluation before administration of any kind of treatment .

The authors explain breast pain -observed prior to surgery by the number of breast biopsies but -especially with taxanes, breast cancer patients can have breast pain ( muscular pain (pectoralis major) and pain due to tumor necrosis in the context of neoadjuvant chemotherapy

Reviewer 2 Report

The AFI is useful to assess the cognitive function of CRCI in breast cancer patients. The paper comprehensively reviewed the interpersonal effectiveness subscale assessment of CRCI.

However, this paper more likely a review styles not an origin one. And

1.  Cited figure 1 should not appear on introduction, Table 1 should not in "method", and Table 7 should be in discussion section rather in result.

2.  The discussion 4-3, about "cytokine" is not related to the results.

3.  The ref. 38 is not the optimal format.

The authors should concise the manuscript and discuss only the results in the current study, and how to apply the self-reported CRCI results here to intervene breast cancer patients.

Round 2

Reviewer 1 Report

I am not satisfied with the answer about breast pain prior to surgery . The authors maintain that  pain  before breast surgery is exclusively related to multiple biopsies and not related to side effects of neoadjuvant chemotherapy .There is more and more evidence that taxanes can induce pain all over the body  . Neurotoxicity is a major concern   described in  many   studies but other types of pain  are also mentioned . This point needs to be discussed again and other references have to be added

Persistent impairments 3 years after (neo)adjuvant chemotherapy for breast cancer:results from the MaTox project

Hans-Jurgen  Hurtz, Hans Tesch, Thomas Gohler, Ulrich Hutzchenreuter, Johanna harde, Lisa Kruggel, Martina Janicke, Norbert Marschner, TMK- group

Breast cancer Res Treat . 2017oct; 165(3):721-731

.Epub2017Jul

Incidence of taxane -induced pain and distress in patients receiving chemotherapy for early-stage breast cancer : a retrospective , outcomes -based survey

S. Saibil, B. Fitzgerald, O.C. Freedman, E.Amir, J. Napolskikh, N.Salvo, G. Dranitsaris, M.Clemons

Curr. Oncol. 2010 Aug; 17(4):42-47doi:10.3747/co.v17i4.562

They explain that there was no tumor necrosis on pathological specimen but chemotherapy  when is efficient- generates  first  tumor necrosis and there after fibrosis. In general  , that is fibrosis which is observed by the pathologist at the time of  breast surgery.

Reviewer 2 Report

No other comment after the revision 

Author Response

We would like to thank you again for the Reviewer's comment.